# Comparative effect of physical exercise versus statins on improving arterial stiffness in patients with high cardiometabolic risk: A network meta-analysis

Iván Cavero-Redondo[1], Jonathan J. Deeks[2], Celia Alvarez-Bueno[1]*, Kate Jolly[2], Alicia Saz-Lara[1], Malcolm Price[2], Carlos Pascual-Morena[1], Vicente Martínez-Vizcaíno[1,3]

1 Universidad de Castilla-La Mancha, Health and Social Research Center, Cuenca, Spain, 2 Institute of Applied Health Research, University of Birmingham, Birmingham, United Kingdom, 3 Universidad Autónoma de Chile, Facultad de Ciencias de la Salud, Talca, Chile

* celia.alvarezbueno@uclm.es

**Data Availability Statement:** All relevant data are within the manuscript and its Supporting Information files.

## Abstract

### Background

The comparative analysis of the effect of several doses of statins against different intensities of physical exercise on arterial stiffness (a measure of cardiovascular risk) could shed light for clinicians on which method is most effective in preventing cardiovascular disease (CVD) and be used to inform shared decision-making between doctors and patients. This study was aimed at analyzing the effect, in high cardiometabolic risk patients, of different statins doses and exercise intensities on arterial stiffness (a measure of cardiovascular risk) by integrating all available direct and indirect evidence in network meta-analyses.

### Methods and findings

We systematically searched MEDLINE, Embase, SPORTDiscus, Cochrane Central Register of Controlled Trials, Cochrane Database of Systematic Reviews, and Web of Science databases from their inception to February 28, 2020; for unpublished trials, we also searched ClinicalTrials.gov. We searched for studies concerning the effect of statins or physical exercise on arterial stiffness, measured by pulse wave velocity (PWV). For methodological quality assessment, Cochrane Collaboration's tool for assessing risk of bias (RoB2) was used. A network geometry graph was used to assess the strength of the evidence. Comparative evaluation of the interventions effect was performed by conducting a standard pairwise meta-analysis and a network meta-analysis (NMA) for direct and indirect comparisons between interventions and control/nonintervention. A total of 22 studies were included in the analyses (18 randomized controlled trials (RCTs) and 4 nonrandomized experimental studies), including 1,307 patients with high cardiometabolic risk from Asia (3 studies), Oceania (2 studies), Europe (10 studies), North America (5 studies), and South America (2 studies). The overall risk of bias assessed with RoB2 was high in all included studies. For

**Funding:** This study was funded by the Consejería de Educación, Cultura y Deportes – Junta de Comunidades de Castilla-La Mancha and European Regional Development Fund (SBPLY/17/ 180501/ 000533 to IC-R, CA-B and VM-V). KJ is part funded by the National Institute for Health Research (NIHR) Applied Research Collaboration West Midlands. The funders had no role in study design, data collection and analysis, decision to publish, or preparation of the manuscript.

**Competing interests:** The authors have declared that no competing interests exist.

**Abbreviations:** ADA, American Diabetes Association; cfPWV, carotid–femoral PWV; CVD, cardiovascular disease; GRADE, Grading of Recommendations, Assessment, Development, and Evaluation; LDLc, low-density lipoprotein cholesterol; MD, mean difference; NMA, network meta-analysis; PRISMA-NMA, Preferred Reporting Items for Systematic Review incorporating Network Meta-Analysis; PWV, pulse wave velocity; RCT, randomized controlled trial; SBP, systolic blood pressure; SD, standard deviation; SUCRA, surface under the cumulative ranking.

standard pairwise meta-analysis and NMA, high-intensity exercise versus control (mean difference (MD) −0.56; 95% CI: −1.01, −0.11; $p = 0.015$ and −0.62; 95% CI: −1.20, −0.04; $p = 0.038$, respectively) and moderate statin dose versus control (MD −0.80, 95% CI: −1.59, −0.01; $p = 0.048$ and −0.73, 95% CI: −1.30, −0.15; $p = 0.014$, respectively) showed significant MDs. When nonrandomized experimental studies were excluded, the effect on high-intensity exercise versus control and moderate statin dose versus was slightly modified. The main limitation of this study was that the magnitude of the effect of the exercise interventions could be underestimated due to regression toward the mean bias because the baseline cardiometabolic risk profile of patients in the physical exercise intervention trials was healthier than those in the statins ones; consequently, more modest improvements in physical exercise interventions compared to statins interventions can be expected. Additionally, we might consider as limitations the small study sizes, the heterogeneous patient groups, the focus on a proxy endpoint (PWV), and the high risk of bias.

## Conclusions

In this NMA, we found that although many patients could benefit from statins for reducing CVD risk, our results support that, considering the beneficial effects of high-intensity exercise on arterial stiffness, it would be worthwhile to refocus our attention on this type of exercise as an effective tool for the prevention of CVD.

## Systematic review registration

PROSPERO CRD42019123120.

## Author summary

### Why was this study done?

- Arterial stiffness is associated with the early stages of vascular aging, being an independent predictor for the onset of acute or chronic cardiovascular disease (CVD).

- Although statins and physical exercise have demonstrated a beneficial role on vascular health, physical exercise has been clearly undervalued in clinical settings.

- There is a lack of clinical trials comparing the effect of statins versus physical exercise on most CVD risk outcomes such as arterial stiffness.

### What did the researchers do and find?

- Using network meta-analysis (NMA), we were able to integrate all available randomized evidence on the effect of statin doses and physical exercise intensities on arterial stiffness in a single analysis for preserving randomization benefits.

- Both moderate statin dose and high-intensity exercise are effective approaches for reducing arterial stiffness.

## What do these findings mean?

- Although many appropriately selected patients could benefit from statins for reducing CVD risk, our result support that, considering the beneficial effects of high-intensity exercise on arterial stiffness, it would be worthwhile to refocus our attention on this type of exercise as an effective tool for the prevention of CVD.

- However, the choice of interventions should be based on the needs and preferences of individual patients.

## Introduction

Arterial stiffness is associated with the early stages of vascular aging [1], being an independent predictor for the onset of acute or chronic cardiovascular disease (CVD) [2]. Inflammation and oxidative stress have been proposed as the underlying mechanisms responsible for the stiffening of vessels' walls, since they induce rapid changes in the endothelium and longer changes in the structural configuration of the arterial wall, through elastin fragmentation, collagen deposition, and smooth muscle cell proliferation [3]. Preventing these deteriorations in vascular structure and function at preclinical stages by using appropriate disease risk stratification strategies leads to health benefits for individuals [4,5]. The accepted gold standard for noninvasive measurement of arterial stiffness is pulse wave velocity (PWV), which has proven to be an independent predictor of cardiovascular events [2,6]. Although PWV is a useful surrogate marker of arterial stiffness with reclassification value over traditional cardiovascular risk estimating scores [2,7,8] and considered as indicative of target organ damage [5], the long-term effects of lowering arterial stiffness (measured by PWV) remain to be demonstrated [5].

In recent years, statins have been the most prescribed drugs worldwide for primary prevention of CVD [9] because of their lipid-lowering effects [10], which could improve the vascular system contributing to reduce arterial stiffness [11,12]. In parallel, physical exercise has demonstrated to be an effective approach for improving arterial stiffness [13]. Although both have demonstrated a beneficial role on vascular health [14], physical exercise has been clearly undervalued in clinical settings. There are many factors why statins are the first line of clinicians' prescription, instead of physical exercise, for primary and secondary prevention of CVD: the ease of prescription, the duration of primary care consultations, and that clinicians have traditionally undervalued the exercise and are rarely trained to prescribe it properly [15,16]. The comparative analysis of the effect of several doses of statins and different intensities of physical exercise on arterial stiffness could shed light for clinicians on the effectiveness of both methods in preventing CVD and be used to inform shared decision-making between doctors and patients.

There is a worldwide concern as to whether the undeniable cardioprotective benefits of the statins have masked the importance of the exercise in the prevention of CVD [17,18]. Moreover, an alleged dilemma about whether to prescribe statins or exercise, because statins would reduce the effect of exercise on fitness, has emerged in recent years [14]. The results provided by the National Runners' and Walkers' Health Study [19] do not suggest that statins reduce the amount of exercise. Likewise, a therapy strategy based on the combination of statins and exercise have proven to be more effective than statin monotherapy not only for improved aerobic capacity, but also in terms of insulin sensitivity and inflammation [20].

However, there is a lack of clinical trials comparing the effect of statins and physical exercise on most CVD risk outcomes such as arterial stiffness, and standard meta-analyses are unable to resolve which treatment is more effective. The network meta-analysis (NMA) approach allows a comprehensive and consistent analysis of all randomized controlled trials (RCTs) comparing, head to head or with placebo, the usually prescribed statins doses and different physical exercise intensities while fully respecting randomization. Thus, this study was aimed at analyzing the effect, in high cardiometabolic risk patients, of different statins doses and exercise intensities on arterial stiffness by integrating all available direct and indirect evidence in NMA.

## Methods

This systematic review and NMA is reported according to the Preferred Reporting Items for Systematic Review incorporating Network Meta-Analysis (PRISMA-NMA) [21] (S1 Table) and the Cochrane Collaboration Handbook [22]. The study protocol was registered in PROS-PERO (registration number: CRD42019123120) and has been published elsewhere [23]. Since in this study the researchers did not collect primary data for the systematic review and NMA, ethical approval was not required.

### Search strategy

Two reviewers (IC-R and CA-B) independently systematically searched MEDLINE, Embase, SPORTDiscus, Cochrane Central Register of Controlled Trials, Cochrane Database of Systematic Reviews, and Web of Science databases from their inception to February 28, 2020; for unpublished trials, we also searched ClinicalTrials.gov. The search strategy combined relevant terms related to (a) cardiometabolic risk; (b) clinical trials; (c) statins or exercise; and (d) arterial stiffness. Finally, the reference lists of the included articles in this review as well as those included in previous systematic reviews and meta-analyses were reviewed for additional relevant studies (S2 Table).

### Eligibility

Studies concerning the effect of statins or physical exercise on arterial stiffness were included. Inclusion criteria were as follows: (1) Type of studies: RCTs or nonrandomized experimental studies and controlled pre–post studies, without language restrictions; (2) Type of participants: studies performed in cardiometabolic disease risk patients as considered by the American Diabetes Association (ADA) [24]. Studies were selected regardless of the age of the participants included. When more than 1 study provided data referring to the same sample, we chose the one presenting the most detailed results or providing the largest sample size; (3) Type of interventions: Studies reporting any type of intervention consisting mainly of statin treatment or physical exercise (endurance, interval training, or combined exercise [either of the previous 2 with resistance]) understood as repeated bouts of physical exercise over time involving more than 1 session/week with a duration of at least 3 weeks were eligible for inclusion. Also, studies comparing different type of statins or comparing different types of physical exercise interventions; (4) Type of outcome assessment: arterial stiffness measured by carotid–femoral PWV (cfPWV). Exclusion criteria were (1) single-arm pre–post studies; (2) studies combining statins or physical exercise with other health interventions, such as nutritional interventions, were excluded when data concerning the effect of statins or physical exercise interventions on arterial stiffness could not be extracted separately; (3) studies where the type and dose of statins or the intensity of physical exercise could not be estimated; and (4) finally, studies reporting

arterial stiffness using other additional PWV measurements sites, such as brachial–ankle PWV or cardiac–ankle PWV (S1 Text).

## Data extraction

After the study selection, 2 reviewers extracted the following information for each study (Tables 1 and 2): (1) year of publication; (2) country; (3) type of study design; (4) sample characteristics (sample size, mean age, and type of population); (5) outcome characteristics (measuring device, baseline cfPWV, and their standard deviation (SD) values as well as arterial stiffness status according with cfPWV reference values) [25]; and (6) intervention characteristics (length, type, and intensity of intervention). According to Cochrane Handbook recommendations, when data on cfPWV SD of change from baseline are lacking, our estimates were calculated on the basis of standard errors, 95% confidence intervals, $p$-values, or t statistics to calculate SD. Finally, when outcomes were scaled inversely, the mean in each group was multiplied by −1.

**Table 1. Characteristics of included studies analyzing the effect of statins on PWV.**

| Study (year) | Country | Study design | Sample size ($n$) | Population characteristics | | Intervention characteristics | | | Outcome | | |
|---|---|---|---|---|---|---|---|---|---|---|---|
| | | | | Age [years (mean ± SD)] | Type of population | Length (weeks) | Type and dose of statin | Level | Measuring device | Basal PWV [m/s (mean ± SD)] | Arterial stiffness status |
| Davenport et al. (2015) | Ireland | Non-RCT | IG1:25 IG2:26 | IG1:66.0 ± 9.5 IG2: 65.5 ± 10.5 | T2DM | 12 and 48 | IG1: Atorvastatin 10 mg IG2: Atorvastatin 80 mg | IG1: Moderate IG2: High | Vicorder | IG1:10.5 ± 1.1 IG2:10.3 ± 1.5 | Normal |
| Fasset et al. (2009) | Australia | RCT | IG:16 CG:18 | IG:62.3 ± 16.3 CG:64.8 ± 15.0 | CKD | 48 | Atorvastatin 10 mg | Moderate | SphygmoCor | IG:8.5 ± 2.2 CG:8.0 ± 1.3 | Normal |
| Grigoropoulou et al. (2019) | Greece | Non-RCT | IG:46 CG:33 | IG:60.0 ± 8.0 CG:59.9 ± 9.0 | T2DM Dyslipidemia | 48 | Atorvastatin 10 mg | Moderate | SphygmoCor | IG:11.3 ± 2.7 CG:10.7 ± 2.4 | Normal |
| Kanaki et al. (2013) | Greece | RCT | IG:25 CG:25 | IG:59.7 ± 8.9 CG:58.8 ± 10.8 | HTA Hypercholesterolemia | 26 | Atorvastatin 10 mg | Moderate | SphygmoCor | IG:11.0 ± 1.8 CG:10.5 ± 2.1 | Elevated |
| Mitsiou et al. (2018) | Greece | RCT | IG:20 IG2:20 | IG1:52.8 ± 8.2 IG2:53.6 ± 8.8 | HTA | 24 | IG1: Rosuvastatin 5 mg IG2: Rosuvastatin 20–40 mg | IG1: Moderate IG2: High | Mobil-O-Graph | IG1:8.4 ± 1.2 IG2:8.2 ± 1.4 | Normal |
| Orr et al. (2009) | USA | RCT | IG:16 CG:10 | IG:53.0 ± 3.0 CG:53.0 ± 2.0 | Overweight/obese | 12 | Atorvastatin 80 mg | High | SPT-301 | IG:11.0 ± 0.4 CG:12.4 ± 0.9 | Elevated |
| Pirro et al. (2006) | Italy | RCT | IG:35 CG:36 | IG:56.0 ± 17.0 CG:58.0 ± 14.0 | Hypercholesterolemia | 4 | Rosuvastatin 10 mg | Moderate | SphygmoCor | IG:9.5 ± 1.9 CG:9.2 ± 2.1 | Normal |
| Raison et al. (2002) | France | RCT | IG:12 CG:11 | IG:56.8 ± 10.9 CG:56.1 ± 9.5 | HTA Hypercholesterolemia | 12 | Atorvastatin 10 mg | Moderate | Complior | IG:12.5 ± 2.7 CG:11.6 ± 1.9 | Elevated |
| Wang et al. (2011) | China | RCT | IG:46 CG:50 | IG:64.2 ± 7.8 CG:65.7 ± 8.2 | HTA | NR | Atorvastatin 20 mg | Moderate | NR | IG:14.4 ± 2.7 CG:14.3 ± 3.1 | Elevated |
| Qi et al. (2008) | China | RCT | IG:45 CG:42 | IG:58.4 ± 5.9 CG:56.4 ± 5.3 | HTA | 12 | Atorvastatin 20 mg | Moderate | Complior | IG:13.1 ± 3.2 CG:12.8 ± 3.1 | Elevated |
| Zhang et al. (2003) | China | RCT | IG:15 CG:15 | IG:63.7 ± 8.8 CG:66.0 ± 9.6 | HTA | 12 | Fluvastatin 40 mg | Low | Complior | IG:12.8 ± 2.5 CG:13.0 ± 3.2 | Normal |

CG, control group; CKD, chronic kidney disease; HTA, hypertension; IG, intervention group; m/s, meters per second; NR, not reported; PWV, pulse wave velocity; RCT, randomized controlled trial; SD, standard deviation; T2DM, type 2 diabetes mellitus.

**Table 2. Characteristics of included studies analyzing the effect of physical exercise on PWV.**

| Study (year) | Country | Study design | Sample size (n) | Population characteristics | | Outcome | | | Intervention characteristics | | |
|---|---|---|---|---|---|---|---|---|---|---|---|
| | | | | Age [years (mean ± SD)] | Type of population | Length (weeks) | Type of physical exercise | Intensity | Measuring device | Basal PWV [m/s (mean ± SD)] | Arterial stiffness status |
| Chrysohoou et al. (2015) | Greece | RCT | IG:50 CG:50 | IG:63.0 ± 9.0 CG:56.0 ± 11.0 | Chronic heart failure | 12 | Interval training | High | NR | IG:9.5 ± 2.5 CG:8.8 ± 1.3 | Normal |
| Dobrosielski et al. (2012) | USA | RCT | IG:70 CG:70 | IG:57.0 ± 6.0 CG:56.0 ± 6.0 | T2DM | 24 | Combined exercise | Moderate | NR | IG:9.2 ± 2.5 CG:9.1 ± 1.7 | Normal |
| Donley et al. (2014) | USA | Non-RCT | IG: 11 CG: 11 | IG:46.0 ± 13.3 CG:44.0 ± 10.0 | Metabolic syndrome | 8 | Endurance training | High | SphygmoCor | IG:7.9 ± 2.0 CG:7.5 ± 1.5 | Normal |
| Guimaraes et al. (2010) | Brazil | RCT | IG1:26 IG2:26 CG:13 | IG1:50.0 ± 8.0 IG2:45.0 ± 9.0 CG:47.0 ± 0.6 | HTA | 16 | IG1: Combined exercise IG2: Combined exercise | IG1: High IG2: High | Complior | IG1:10.2 ± 1.7 IG2:9.4 ± 0.9 CG:10.2 ± 1.8 | Elevated |
| Koh et al. (2010) | Australia | RCT | IG1:13 IG2:14 CG:15 | IG1:52.3 ± 10.9 IG2:52.1 ± 13.6 CG:51.3 ± 14.4 | CKD | 24 | IG1: Endurance training IG2: Endurance training | IG1: Moderate IG2: Moderate | SPT-301 | IG1:9.1 ± 2.8 IG2:9.7 ± 3.2 CG:8.7 ± 2.5 | Normal |
| Loimaala et al. (2009) | Finland | RCT | IG:25 CG:25 | IG:53.6 ± 6.2 CG:54.0 ± 5.0 | Type 2 diabetes | 24 | Combined exercise | High | CircMon B 202 | IG:14.1 ± 2.5 CG:14.1 ± 2.5 | Elevated |
| Madden et al. (2013) | Canada | RCT | IG:25 CG:27 | IG:68.5 ± 0.9 CG:70.0 ± 0.8 | T2DM HTA Hyperlipidemia | 24 | Endurance training | High | Complior | IG:13.4 ± 0.7 CG:12.0 ± 0.6 | Normal |
| Mora-Rodriguez et al. (2017) | Spain | RCT | IG:25 CG:25 | 53.5 ± 8.9 | Metabolic syndrome Hypertension | 24 | Interval training | High | SphygmoCor | IG:8.5 ± 2.1 CG: 8.5 ± 2.2 | Normal |
| Nualnim et al. (2012) | USA | Non-RCT | IG:24 CG:19 | IG:58.0 ± 9.8 CG:61.0 ± 8.6 | Pre-HTA HTA | 12 | Endurance training | Moderate | VP-2000 | IG:9.1 ± 1.0 CG:9.4 ± 0.7 | Normal |
| e Silva et al. (2019) | Brazil | RCT | IG:15 CG:15 | IG:50.0 ± 17.2 CG:58.0 ± 15.0 | CKD | 16 | Endurance training | Moderate | SphygmoCor | IG:8.5 ± 2.9 CG:10.3 ± 3.8 | Normal |
| Slivovskaja et al. (2018) | Lithuania | RCT | IG:84 CG:42 | IG:53.9 ± 6.4 CG:52.0 ± 7.7 | Metabolic syndrome | 8 | Endurance training | High | SphygmoCor | IG:8.5 ± 1.4 CG:8.0 ± 1.1 | Normal |

CG, control group; CKD, chronic kidney disease; HTA, hypertension; IG, intervention group; m/s, meters per second; NR, not reported; PWV, pulse wave velocity; RCT, randomized controlled trial; SD, standard deviation; T2DM, type 2 diabetes mellitus.

## Categorization of the interventions' available evidence

Statin interventions were classified by type and dose as high, moderate, and low following the Statin Dosing and ACC/AHA Classification of Intensity [26]. Furthermore, physical exercise interventions, considered as a subset of structured and repetitive physical activity, were classified by intensity as high, moderate, or light following the American College of Sports Medicine guidelines for prescribing exercise [27]. This classification of statin doses and exercise intensity was performed by 2 researchers (IC-R and CA-B) independently.

## Risk of bias assessment

Two researchers (IC-R and CA-B) independently conducted a methodological quality assessment of the included RCTs by applying the Cochrane Collaboration's tool for assessing risk of bias (RoB2) [28]. according to the Cochrane Collaboration Handbook recommendations [22]. Risk of bias was evaluated according to 6 domains: selection bias, performance bias, detection

bias, attrition bias, reporting bias, and other bias. Overall, bias was categorized as "low risk of bias" if the paper had been classified as "low risk" in all domains, "some concerns" if there was at least 1 domain with rating of "some concern," and "high risk of bias" if there was at least 1 domain with a "high risk," or several domains with some concerns in such a way that the validity of the results could be affected. As in studies that include physical exercise interventions, patient allocation to treatment could not be blinded, thus patient blinding domain was deemed as a high risk of bias, and we did not include this domain in the overall risk of bias assessment. Any disagreements were solved by discussion and a third reviewer (VM-V).

## Grading the quality of evidence

The Grading of Recommendations, Assessment, Development, and Evaluation (GRADE) tool was used to evaluate the quality of the evidence and make recommendations [29]. Each outcome obtained a high, moderate, low, or very low evidence value, depending on the design of the studies, risk of bias, inconsistency, indirect evidence, imprecision, and publication bias.

## Data synthesis and statistical analysis

The included clinical trials were summarized qualitatively in an ad hoc table describing the types of direct and indirect comparisons. As noted above, we conducted our NMA accordingly with PRISMA-NMA statement distinguishing the following steps:

- A network geometry graph was used to display the evidence in the network. In this graph, the size of the nodes was proportional to the number of participants in trials who received the intervention specified in the node, the thickness of continuous line connecting nodes proportional to number of participants in trials directly comparing the 2 treatments [30].

- Consistence assessment, by checking whether the treatment effects estimated from direct comparisons were consistent with those estimated by indirect comparisons procedures. For this aim, we used the Wald test; moreover, we assessed local inconsistency using the side-splitting method.

- Comparative evaluation of the interventions effect, by conducting a standard pairwise meta-analysis for comparisons between interventions and control/nonintervention. For this, we used the random effects DerSimonian–Laird method [31], and the statistical heterogeneity was examined by calculating the $I^2$ statistic, separately for each statin doses and for each exercise intensity, which ranges from 0% to 100%. According to the values of $I^2$, the heterogeneity will be considered as not important (0% to 40%), moderate (30% to 60%), substantial (50% to 90%), or considerable (75% to 100%) [22]. Additionally, the corresponding $p$-values were also be considered. Finally, to determine the size and clinical relevance of heterogeneity, the $\tau^2$ statistic was calculated. A $\tau^2$ estimate of 0.04 may be interpreted as a low, 0.14 as a moderate, and 0.40 as a substantial degree of clinical relevance of heterogeneity [32]. These results were displayed by creating both forest plots and a league table.

- Transitivity assessment, to check that the synthesis of direct comparisons of 2 treatments have been conducted in studies that were similar in the most important clinical and methodological characteristics; thus, it supposed to assume that the populations included in these studies were similar in the baseline distribution of the effect modifier. For this aim, we checked that all the participants in the studies included in the NMA have the same baseline characteristics (on average) that might modify the treatment effect [33].

- Relative rankings of treatments. Once we had comparatively estimated the effectiveness of the different treatment strategies, the next step was to rank the treatments in order to

identify superiority. The probability that each intervention, statin, or physical exercise were the most effective was presented graphically using rankograms [34]. Additionally, the surface under the cumulative ranking (SUCRA) was estimated for each intervention. SUCRA involves the assignment of a numerical value between 0 and 1 to simplify the classification of each intervention in the rankogram. The best intervention would obtain a value for SUCRA close to 1, and the worst intervention would be a value close to 0 [30].

- Sensitivity analysis and small-study effect. Sensitivity analyses were conducted to assess the robustness of the summary estimates and to detect whether any particular study represented a large proportion of the heterogeneity. Additionally, a sensitivity analysis was performed excluding nonrandomized experimental studies and controlled pre–post studies. For examining the presence of bias due to small-study effect, a network funnel plot was used to visually scrutinize the criterion of symmetry [35]. All the analyses were conducted in Stata 15.0 (Stata, College Station, Texas, USA).

## Results

A total of 22 studies [36–57] (18 RCTs and 4 nonrandomized experimental studies) were included in the analyses (Fig 1). Moreover, 11 studies analyzed the effect of statins and 11 the effect of physical exercises on cfPWv (Tables 1 and 2). Two of the physical exercise studies have 3 arms (2 interventions and 1 control). The statin therapy includes atorvastatin, rosuvastatin, and fluvastatin. In most studies, the intervention group received a moderate dose of statin (9 studies). Low statin dose was evaluated only in 1 study. High-intensity exercise was evaluated in 7 studies (8 intervention samples), and moderate-intensity exercise was evaluated in 4 studies (5 intervention samples) (Fig 2).

### Transitivity assessment

Statin and physical exercise studies included a similar total number of patients (587 and 720, respectively). Participants in physical exercise interventions groups had lower baseline cardiometabolic risk profile than statins intervention group: systolic blood pressure (SBP) 133.0 mmHg versus 140.8 mmHg ($p = 0.023$), total cholesterol 5.0 mmol/L versus 6.0 mmol/L ($p = 0.001$), and low-density lipoprotein cholesterol (LDLc) 3.1 mmol/L versus 4.2 mmol/L ($p = 0.001$) (S3 Table).

### Risk of bias

The overall risk of bias was high in all included studies. Regarding each domain, the risk of bias for deviations from intended interventions domain was high in all studies; the risk of bias for statins and exercise was high for, respectively, 100% and 90.9% of studies for deviations from intended interventions. However, the risk of bias for statins and exercise was low for, respectively, 90.9% and 72.7% of studies for measurement of the outcome domain, and 90.9% and 90.9% for missing outcome data domain (S1 and S2 Figs).

When the quality of evidence of each pairwise comparison was evaluated using the GRADE system, 50.0% of the pairwise comparisons were categorized as high, 33.3% as moderate, and 16.7% as very low (S4 Table).

### Arterial stiffness

In Table 3, considering both the direct pairwise pooled estimates (upper diagonal) and the NMA estimates (lower diagonal), the highest mean differences (MDs) in cfPWv were for high

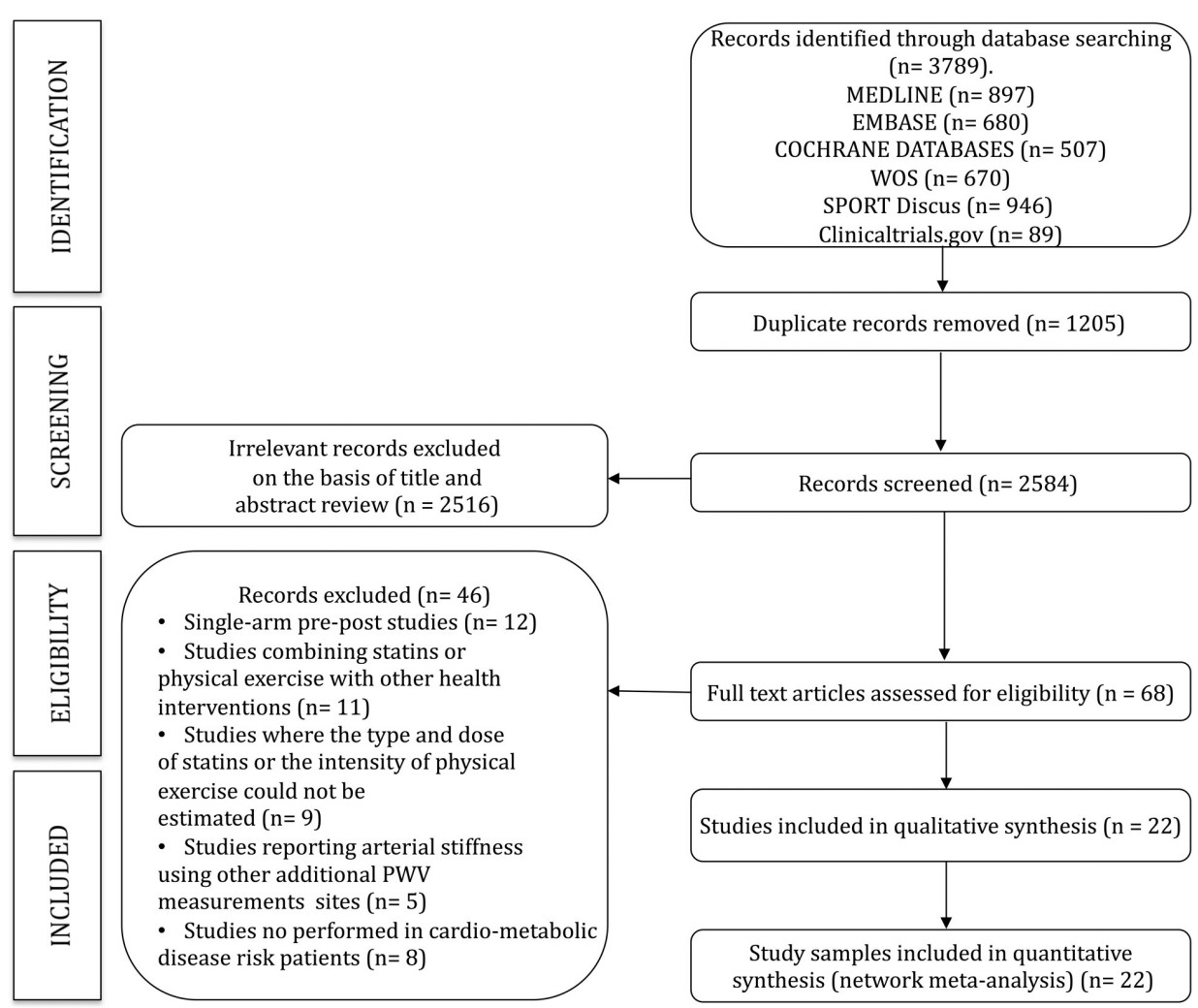

**Fig 1. PRISMA flowchart.** PRISMA, Preferred Reporting Items for Systematic Reviews; PWV, pulse wave velocity; WOS, Web of Science.

statin dose versus control (−1.17; 95% CI: −3.50, 1.16; $p$ = 0.326 and −0.86; 95% CI: −1.82, 0.10; $p$ = 0.080, respectively). Furthermore, high-intensity exercise versus control (−0.56; 95% CI: −1.01, −0.11; $p$ = 0.015 and −0.62; 95% CI: −1.20, −0.04; $p$ = 0.038, respectively) and moderate statin dose versus control (−0.80, 95% CI: −1.59, −0.01; $p$ = 0.048 and −0.73, 95% CI: −1.30, −0.15; $p$ = 0.014, respectively) had confidence intervals which excluded the no effect value.

## Treatment ranking

The high statin dose showed the higher SUCRA (74%) (Fig 3, S5 Table, S3 Fig). Moderate statin dose and high-intensity exercise showed the second and third higher SUCRA (67% and 60%), showing high-intensity exercise as the best mean and median rank (lower mean and median values mean better rank).

## Sensitivity analysis, heterogeneity, and small-study effect

The pooled MD was not significantly modified when the individual study data were removed, one at a time, from any pairwise comparison analysis. When nonrandomized experimental

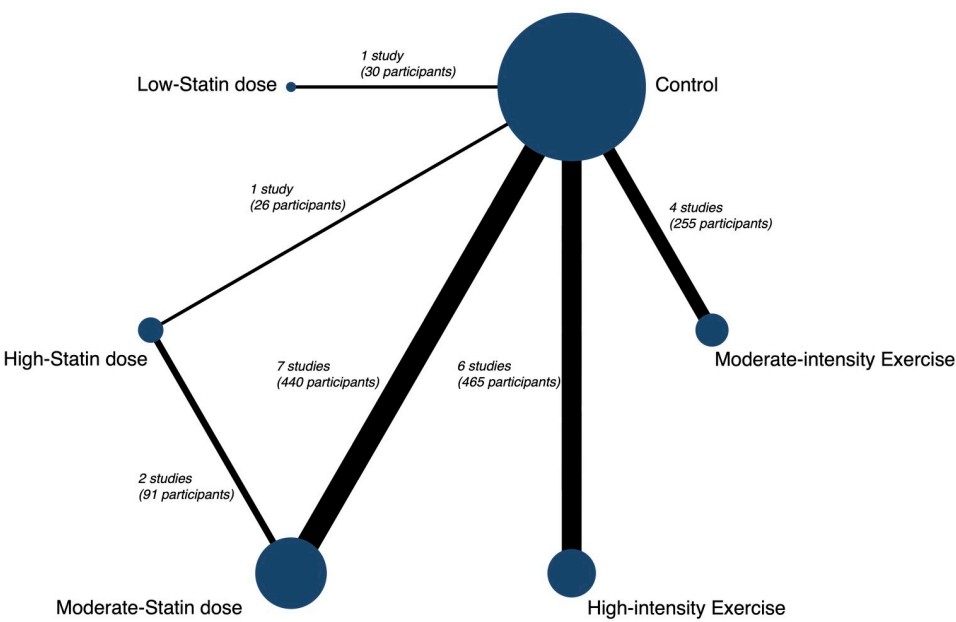

**Fig 2. Network of available comparisons between statins doses and physical exercise intensities interventions in arterial stiffness.** Size of node is proportional to number of trial participants, and thickness of continuous line connecting nodes is proportional to number of participants randomized in trials directly comparing the 2 treatments. Dash lines display indirect comparisons.

studies and controlled pre–post studies were excluded from the pairwise comparison analysis, the effect on moderate statin dose versus control (−0.52; 95% CI: −0.97, −0.08) and high-intensity exercise versus control (−0.55; 95% CI: −1.01, −0.08) was slightly modified. Furthermore, moderate statin dose showed substantial heterogeneity ($I^2 = 61.2$, $\tau^2 = 0.650$). The other direct comparison showed no heterogeneity ($I^2 = 0.0$, $\tau^2 = 0.000$) (S6 Table). Finally, there was evidence of small-study effect in funnel plot asymmetry and Egger test for high-intensity exercise versus control ($p = 0.035$), but not for all other comparisons: moderate statin dose versus

**Table 3. Pooled MDs on arterial stiffness.**

| Placebo | −1.17 (−3.50, 1.16) | −0.80 (−1.59, −0.01) | −0.50 (−3.20, 2.20) | −0.27 (−1.00, 0.46) | −0.56 (−1.01, −0.11) |
|---|---|---|---|---|---|
| −0.86 (−1.82, 0.10) | High statin dose | 0.11 (−0.59, 0.80) | na | na | na |
| **−0.73** **(−1.30, −0.15)** | 0.13 (−0.67, 0.94) | Moderate statin dose | na | na | na |
| −0.50 (−3.29, 2.29) | 0.38 (−2.59, 3.31) | 0.23 (−2.62, 3.07) | Low statin dose | na | na |
| −0.28 (−1.16, 0.60) | 0.58 (−0.72, 1.88) | 0.45 (0.60, −1.46) | 0.22 (−2.70, 3.14) | Moderate-intensity exercise | na |
| **−0.62** **(−1.20, −0.04)** | 0.24 (−0.85, 1.34) | 0.11 (−0.67, 0.89) | −0.12 (−2.96, 2.73) | −0.34 (−1.39, 0.71) | High-intensity exercise |

Upper right triangle gives the pooled MDs from pairwise comparisons (column intervention relative to row), and lower left triangle gives the pooled MDs from the NMA (row intervention relative to column).

MD, mean difference; na, not available; NMA, network meta-analysis.

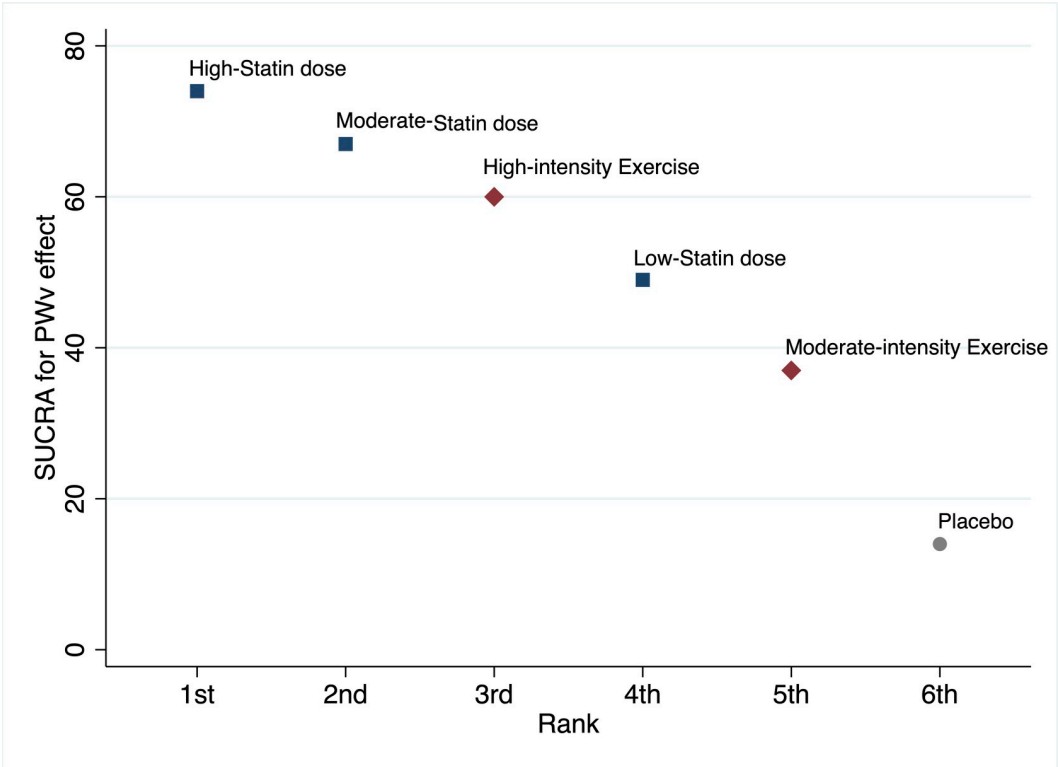

**Fig 3. SUCRA.** PWV, pulse wave velocity; SUCRA, surface under the cumulative ranking.

control ($p = 0.294$), moderate statin dose versus high statin dose ($p = 0.686$), and moderate-intensity exercise versus control ($p = 0.557$) (S4 Fig).

## Discussion

Statins as pharmacological treatment and physical exercise as lifestyle recommendation are the most used strategies for the treatment of patients with high cardiometabolic risk. Clearly, as for any treatment, doing more harm than good with this class of treatments should be avoided (primum non nocere). Taking this into account, we presented arterial stiffness as an outcome of vascular health in which an effective treatment could be associated with a decrease in risk of CVD. In this NMA of the effect of 3 statin doses and 2 intensities of physical exercise on arterial stiffness, moderate statin dose and high-intensity exercise seemed to be more effective. The effect on arterial stiffness, as measured by cfPWv, of statins and physical exercise varied considerably depending on the doses and intensities, respectively.

Our findings on the available scientific evidence allow prescribers and their patients an understanding of the clinical circumstances where statins might provide substantial improvements in arterial stiffness, and the intensity of exercise required that could yield potentially similar gains. Our analysis covered 1,307 cardiometabolic disease/risk patients (587 for statins and 720 for physical exercise), but only a limited number of studies examined the effect of high and low statin doses as well as for light-intensity exercise. Even so, with the studies included, a dose response trend can be assumed both for statins and exercise. This is evident from the SUCRA findings that showed that the high statin dose may be the best treatment, but the limited number of studies may not find a statistically significant effect, but it should be highlighted that when choosing the safest treatment, increasing the dose of statins has been

associated with pain and muscle damage, liver injury, or increased risk of type 2 diabetes [58,59]. Conversely, exercise has demonstrated positive benefits for most prevalent noncommunicable diseases [60].

The findings of our NMA suggest that moderate statin dose and high-intensity exercise interventions are often potentially effective in terms of improvements on arterial stiffness. High-intensity exercise interventions should therefore be considered as a viable alternative to, or alongside, moderate statin dose therapy. Indeed, an increasing number of experts recommend prescribing an "exercise pill" as a preventive strategy to reduce CVD [61,62]. However, as opposed to the findings of this NMA, international guidelines have reduced the threshold for statin treatment after intensive lifestyle modifications and considerably extended both the scope and dosage of statin treatment [63].

We used a comprehensive search strategy and searched pertinent sources to retrieve potentially eligible RCTs. It therefore seems unlikely that we missed any relevant trial. Using NMA, we were able to integrate all available randomized evidence on the effect of statin doses and physical exercise intensities on arterial stiffness in a single analysis for preserving randomization benefits. The integration of direct and indirect comparisons results in a gain of statistical power for formal comparisons of statins doses and physical exercise intensities with placebo [12,13].

The results of this NMA may be affected by some limitations that should be acknowledged. First, the high overall risk of bias in all included studies, mainly due to all trials lacked information regarding deviations from intended interventions domain from the RoB2; however, based on the others quality domains of the RoB2, the methodological quality of included trials was generally satisfactory. Second, the inclusion of nonrandomized experimental studies and controlled pre–post studies, although the exclusion of the nonrandomized experimental studies in the sensitivity analysis was only reflected in small modifications of the effect in both moderate statin dose versus control and high-intensity exercise versus control. Third, the limited number of samples included in the evaluation of some interventions (high statin dose versus control $n = 1$ and low statin dose versus control $n = 1$), which influenced the GRADE evaluation. Fourth, the transitivity assessment showed a lower risk cardiometabolic profile at baseline in the patients in physical exercise interventions trials than those in statins interventions; consequently, more modest improvements in physical exercise interventions compared to statins interventions can be expected. Fifth, a high heterogeneity in the characteristics of population included in the studies such as clinical (including patients with hypertension, obesity, diabetes mellitus, chronic kidney disease, or heart failure) or cardiometametabolic risk profile; certainly, this heterogeneity in cardiovascular risk status affects the statistical procedures of the study, and their estimates must be cautiously examined, although it also makes our findings generalizable to real clinical practice. Sixth, an additional source of heterogeneity is that the devices used for measuring cfPWv differ largely, since while some used a tonometric method, other were oscillometric devices, but the robustness of our estimates was not modified after removing studies including oscillometric devices in the sensitivy analyses. Seventh, there was evidence of small-study effect on the funnel plot asymmetry (S4 Fig) and Egger test, for high-intensity exercise versus control comparison, produced mainly by the scarcity of studies and their small sample sizes. Furthermore, the funnel plot asymmetry showed that the studies appear to be highly clustered round common estimates for each type of intervention; this phenomenon can be explained by the similarity of sample sizes, but in addition to the reliability of the direction and the power of the effect for each type of intervention. Eighth, it should be acknowledged that because in this NMA, comparisons between physical exercise and statins are indirect, the same biases may occur as in nonrandom comparisons, and the results should be carefully interpreted. Finally, our results are based on the analysis of the effect of statins and

exercise on reducing PWV, and although consistent evidence support that PWV is a good predictor of CVD, it is indeed a surrogate measure of the appropriate main outcomes for these interventions, which are CVD events, CVD mortality, and all-cause mortality, with all the clinical and epidemiological implications that this fact entails.

## Conclusions

In summary, our study confirms that both statins and physical exercise are effective approaches for reducing arterial stiffness. Our results, based on data from experimental studies, represent the best available evidence of the effect on arterial stiffness of these 2 different therapeutic and preventive strategies. Although many appropriately selected patients could benefit from the effects of statins on arterial stiffness, our results support that, considering the beneficial effects of high-intensity exercise on arterial stiffness, the choice of this intervention should be based on the needs and preferences of each patient.

## Disclaimer

The views expressed in this article are those of the author(s) and not necessarily those of the NHS, the NIHR, or the Department of Health and Social Care.

## Supporting information

**S1 Table. PRISMA NMA Checklist of Items to Include When Reporting A Systematic Review Involving a Network Meta-analysis.**
(DOCX)

**S2 Table. Search strategy for the MEDLINE database.**
(DOCX)

**S3 Table. Pooled baseline characteristics from statins and physical exercise interventions.**
(DOCX)

**S4 Table. Quality grading of evidence.**
(DOCX)

**S5 Table. Effectiveness ranking of stain doses and physical exercise intensities interventions.**
(DOCX)

**S6 Table. Heterogeneity statistics for each pairwise comparison.**
(DOCX)

**S1 Fig. Risk of bias for statin interventions.**
(DOCX)

**S2 Fig. Risk of bias for physical exercise interventions.**
(DOCX)

**S3 Fig. Rankogram for each intervention.**
(DOCX)

**S4 Fig. Funnel plot for comparison-specific pooled mean differences.**
(DOCX)

**S1 Text. References of excluded studies (with reasons) from network meta-analysis.**
(DOCX)

## Author Contributions

**Conceptualization:** Iván Cavero-Redondo, Jonathan J. Deeks, Kate Jolly, Vicente Martínez-Vizcaíno.

**Formal analysis:** Iván Cavero-Redondo, Celia Alvarez-Bueno, Alicia Saz-Lara.

**Investigation:** Jonathan J. Deeks, Celia Alvarez-Bueno.

**Methodology:** Iván Cavero-Redondo, Jonathan J. Deeks, Celia Alvarez-Bueno, Carlos Pascual-Morena, Vicente Martínez-Vizcaíno.

**Supervision:** Jonathan J. Deeks, Celia Alvarez-Bueno, Vicente Martínez-Vizcaíno.

**Validation:** Alicia Saz-Lara.

**Visualization:** Malcolm Price.

**Writing – original draft:** Iván Cavero-Redondo.

**Writing – review & editing:** Jonathan J. Deeks, Celia Alvarez-Bueno, Kate Jolly, Alicia Saz-Lara, Malcolm Price, Carlos Pascual-Morena, Vicente Martínez-Vizcaíno.

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
