## [Editor Report · Decision Letter 0]

6 Jul 2020

Dear Dr Alvarez-Bueno, 

Thank you for submitting your manuscript entitled "Comparative effect of physical exercise versus statins on improving arterial stiffness in patients with high cardiometabolic risk: a network meta-analysis" for consideration by PLOS Medicine.

Your manuscript has now been evaluated by the PLOS Medicine editorial staff and I am writing to let you know that we would like to send your submission out for external peer review.

Kind regards,

Helen Howard, for Clare Stone PhD 

Acting Editor-in-Chief

PLOS Medicine 

plosmedicine.org

---

## [Decision Letter · Decision Letter 1]

28 Sep 2020

Dear Dr. Alvarez-Bueno,

Thank you very much for submitting your manuscript "Comparative effect of physical exercise versus statins on improving arterial stiffness in patients with high cardiometabolic risk: a network meta-analysis" (PMEDICINE-D-20-03097R1) for consideration at PLOS Medicine. 

Your paper was evaluated by a senior editor and discussed among all the editors here. It was also evaluated by three independent reviewers, including a statistical reviewer. The reviews are appended at the bottom of this email and any accompanying reviewer attachments can be seen via the link below:

[LINK]

In light of these reviews, I am afraid that we will not be able to accept the manuscript for publication in the journal in its current form, but we would like to consider a revised version that addresses the reviewers' and editors' comments. Obviously we cannot make any decision about publication until we have seen the revised manuscript and your response, and we plan to seek re-review by one or more of the reviewers. 

We expect to receive your revised manuscript by Oct 19 2020 11:59PM. Please email us (plosmedicine@plos.org) if you have any questions or concerns.

We look forward to receiving your revised manuscript. 

Sincerely,

Emma Veitch, PhD

PLOS Medicine

On behalf of Adya Misra, PhD, Senior Editor, 

PLOS Medicine

plosmedicine.org

*In the Abstract (ideally Methods and Findings section), please include a brief note summarising any key limitation(s) of the study's methodology.

*The main Limitations section of the paper seems to omit mention of a key point, that is the focus specifically in this network meta on a surrogate outcome measure - some discussion of this could be added.

*The authors have used the PRISMA guideline (specifically the one focussed on network meta-analyses) to help with reporting, and this is valuable, but a "flowchart" of study eligibility/inclusion seems to have been omitted - please see the generic PRISMA statement (https://www.equator-network.org/reporting-guidelines/prisma/) for a template Word figure for the flowchart and that could be filled out and included as a figure with the revised paper. 

Comments from the reviewers:

Reviewer #1: See attachment

Michael Dewey

Reviewer #2: 

The Authors of this paper carried out a network meta-analysis on the effect of different statins doses and exercise intensities on arterial stiffness (expressed as carotid-femoral pulse wave velocity), in high cardio-metabolic risk patients. Nineteen studies were included in the analyses, including 1151 patients.

The main results showed in addition to a benefit from statins for reducing cardiovascular risk, also a beneficial effects of high-intensity exercise on arterial stiffness. 

This is an interesting paper, however there are some concerns.

1. The results may be affected by some limitations: a) the high overall risk of bias in all included studies; b) the inclusion of non-randomized controlled trials; c) the limited number of samples included in the evaluation of some interventions (high-statin dose vs control n=1, low-statin dose vs control n=1), that also affected the GRADE evaluation; d) the statistical difference in baseline cardio-metabolic risk profile between physical exercise and statin intervention group; e) a high heterogeneity in the characteristics of the studies (e.g. cardio-metabolic risk profile of the populations, arterial stiffness assessment).

2. The visual inspection of the funnel plot pointed out a slight asymmetrical distribution. Moreover, the result of Egger's test was not reported in the text. 

3. Please consider another reference for the effect of statins on arterial stiffness, expressed only as carotid-femoral pulse wave velocity (Clin Exp Hypertens. 2018;40(7):601-608. doi: 10.1080/10641963.2017.1411498), because Upala et al (ref 11) combined the results of different expression of arterial stiffness.

4. Please check typo-errors at page 4,9, and Figure 2

5. Please re-check the references (e.g. reference 12, it should be Upala et al)

6. The different number of samples/studies included reported in the abstract and manuscript may generate confusion

Reviewer #3: 

Although many patients could benefit from statins to reduce CVD risk, our results

support that, considering the beneficial effects of high-intensity exercise on arterial

stiffness, it would be worthwhile to refocus our attention on this type of exercise as an

effective tool for prevention of CVD.

A well written paper with practical implications.

Any issues with the original submission were resolved in the revision in a satisfactory way.

[LINK]

---

## [Decision Letter · Decision Letter 2]

15 Jan 2021

Dear Dr. Alvarez-Bueno,

Thank you very much for re-submitting your manuscript "Comparative effect of physical exercise versus statins on improving arterial stiffness in patients with high cardiometabolic risk: a network meta-analysis" (PMEDICINE-D-20-03097R2) for review by PLOS Medicine.

I have discussed the paper with my colleagues and the academic editor and it was also seen again by two reviewers. I am pleased to say that provided the remaining editorial and production issues are dealt with we are planning to accept the paper for publication in the journal.

[LINK]

We look forward to receiving the revised manuscript by Jan 22 2021 11:59PM.   

Sincerely,

Artur Arikainen, 

Associate Editor 

PLOS Medicine

plosmedicine.org

Requests from Editors:

1. Financial Disclosure: Please add “The funders had no role in study design, data collection and analysis, decision to publish, or preparation of the manuscript.”; or explain otherwise.

2. Data Availability Statement (DAS): If the data are owned by a third party but freely available upon request, please note this and state the owner of the data set and contact information for data requests (web or email address). Note that a study author cannot be the contact person for the data. If the data are not freely available, please describe briefly the ethical, legal, or contractual restriction that prevents you from sharing it. Please also include an appropriate contact (web or email address) for inquiries (again, this cannot be a study author).

3. Please include line numbers in the margin throughout.

4. Abstract:

a. Please include search dates, and databases searched.

b. Please include a summary of included study types (RCTs, non-randomized cohorts, etc.), and a summary of study location by global region/country.

c. Please include the name of the quality/bias assessment method, and a summary of included study quality.

d. Please include p values alongside 95% CIs/CrIs for comparisons.

e. Please mention the results of the sensitivity analysis (removal of non-randomized studies).

f. Please clarify where in the main Results/Tables the following data values can be verified (we were unable to locate all of these): “For standard pairwise meta-analysis and a network meta-analysis, high-intensity exercise versus control (mean difference (MD) -0.82; 95%CI: -1.48, -0.15; and -0.80; 95%CrI: -1.55, -0.06, respectively) and moderate-statin dose versus control (MD -0.80, 95%CrI: -1.59, -0.01; and -0.74, 95%CrI: -1.33, -0.15, respectively) showed significant mean differences.”

g. At the end of the ‘Methods and findings’ subsection, please more clearly state: “Limitations of this study were…”; and mention other major issues, such as small study sizes, heterogeneous patient groups, focus on a proxy endpoint, and high risk of bias.

h. Conclusions: Please begin with: “In this network meta-analysis, we found that…”

5. Author Summary: Under "What do these findings mean", please mention that choices of interventions will need to be based on the needs and preferences of individual patients (i.e., not everyone can do high-intensity exercise, and statins have their drawbacks too).

6. Please use the "Vancouver" style for reference formatting, and see our website for other reference guidelines https://journals.plos.org/plosmedicine/s/submission-guidelines#loc-references. Citations should be in square brackets, before punctuation, and not superscript, eg. “…[1,2].”

7. Methods:

a. Please include the exact search date (including day).

b. Please mention that separate ethical approval was not required for your study.

8. Results: Please include p values alongside 95% CIs/CrIs for all comparisons.

9. Please remove the Contributors, Funding, and Disclosures sections on p. 16-17 – these details are taken from the online submission from.

10. Please provide access details (eg. volume/issue/pages, DOI or URL) for references 9, 22, 26.

11. Please upload each item of Supporting Information as a separate file.

12. When completing the PRISMA checklist, please use section and paragraph numbers, rather than page numbers.

---

Comments from Reviewers:

Reviewer #1: The authors have addressed all my points

Michael Dewey

Reviewer #2: The paper was improved. No further comments.

[LINK]

---

## [Editor Report · Decision Letter 3]

19 Jan 2021

Dear Dr. Alvarez-Bueno,

Thank you very much for re-submitting your manuscript "Comparative effect of physical exercise versus statins on improving arterial stiffness in patients with high cardiometabolic risk: a network meta-analysis" (PMEDICINE-D-20-03097R3) for review by PLOS Medicine.

I am pleased to say that provided the remaining minor editorial and production issues are dealt with we are planning to accept the paper for publication in the journal.

[LINK]

We look forward to receiving the revised manuscript by Jan 26 2021 11:59PM.   

Sincerely,

Artur Arikainen, 

Associate Editor 

PLOS Medicine

plosmedicine.org

Requests from Editors:

1. Abstract: 

a. Line 47: Please give numbers of studies included from each region.

b. Line 55: Delete duplicate “The main”

2. Some final additional comments from the Academic Editor for consideration (apologies that these were not included in our previous email): 

a. Methods/interpretation: The authors should acknowledge and discuss that network meta-analysis (in particular when comparisons are indirect) is by many considered as prone to the same issues that non-randomised comparisons are subject to. So, all indirect comparisons between PA and statins are potentially biased and should be carefully interpreted.

b. Intro/discussion: I recommend that the authors tone down the PA vs statin debate and recommendations. It is not really one or the other but the emphasis should be on both (and this paper arguing that we should put more emphasis on PA - but not at the cost of drug treatment).

c. Discussion: Following from the point above, one key limitation of the work is that focuses on one surrogate outcome but then draws far-reaching conclusions on clinical and public health decision making. This is stretching things too far and should be emphasised.

3. Line 369: Typo in “meta-analysis”.

4. Lines 414-420: Please remove the Data Availability and financial Disclosures sections – these details are taken from the online submission form.

5. Reference 63: Please remove italic formatting here.

6. When completing the PRISMA checklist, please use section and paragraph numbers, rather than page or line numbers – page and line numbers will change in the final published document.

---

Comments from Reviewers:

n/a

[LINK]

---

## [Editor Report · Decision Letter 4]

22 Jan 2021

Dear Dr Alvarez-Bueno, 

On behalf of my colleagues and the Academic Editor, Kazem Rahimi, I am pleased to inform you that we have agreed to publish your manuscript "Comparative effect of physical exercise versus statins on improving arterial stiffness in patients with high cardiometabolic risk: a network meta-analysis" (PMEDICINE-D-20-03097R4) in PLOS Medicine.

PRESS

Sincerely, 

Artur A. Arikainen 

Associate Editor 

PLOS Medicine